# An Emotion and Attention Recognition System to Classify the Level of Engagement to a Video Conversation by Participants in Real Time Using Machine Learning Models and Utilizing a Neural Accelerator Chip

**Janith Kodithuwakku *** , **Dilki Dandeniya Arachchi and Jay Rajasekera ***

Digital Business and Innovations, Tokyo International University, Saitama 350-1197, Japan; dadmahindika@gmail.com
* Correspondence: mjpkodithuwakku@gmail.com (J.K.); jrr@tiu.ac.jp (J.R.)

**Abstract:** It is not an easy task for organizers to observe the engagement level of a video meeting audience. This research was conducted to build an intelligent system to enhance the experience of video conversations such as virtual meetings and online classrooms using convolutional neural network (CNN)- and support vector machine (SVM)-based machine learning models to classify the emotional states and the attention level of the participants to a video conversation. This application visualizes their attention and emotion analytics in a meaningful manner. This proposed system provides an artificial intelligence (AI)-powered analytics system with optimized machine learning models to monitor the audience and prepare insightful reports on the basis of participants' facial features throughout the video conversation. One of the main objectives of this research is to utilize the neural accelerator chip to enhance emotion and attention detection tasks. A custom CNN developed by Gyrfalcon Technology Inc (GTI) named GnetDet was used in this system to run the trained model on their GTI Lightspeeur 2803 neural accelerator chip.

**Keywords:** artificial intelligence (AI); convolutional neural network (CNN); facial emotion recognition (FER); support vector machine (SVM)

## 1. Introduction

Video conferencing has increased dramatically, especially because of the pandemic situation. Video conferencing technologies such as Zoom, Microsoft Teams, Skype, and Google Meets have received huge popularity because of their increased usage over recent years. Schools and businesses use these tools to enable them to continue online teaching and working from home in pandemic times. Even in pre-pandemic times, distance education (distance learning) was becoming popular among students [1].

The use of online conferencing tools created a new need for monitoring participants for their engagement in a class or a meeting. This is an exceedingly difficult objective to achieve by teachers or business managers. For example, students in an online lecture can focus on something other than the lecture but they can pretend to attend the meeting by just using the tool. Thus, getting a better idea of how well those students pay attention is not an easy task in a scenario such as that depicted in Figure 1b. As a solution, some teachers ask their students to turn on their web cameras when they attend a lecture, but this is only feasible with a small audience. For a big class, it is impossible for the lecturer to look at every participant while teaching. Most importantly, checking only once or twice throughout the meeting is not enough. There is a need for a system that can continuously look at participants throughout the meeting and generate a report on the level of engagement of each student/participant on the basis of their facial features. Such a system will be very useful for the teacher/organizer for evaluating participants' attendance to a meeting.

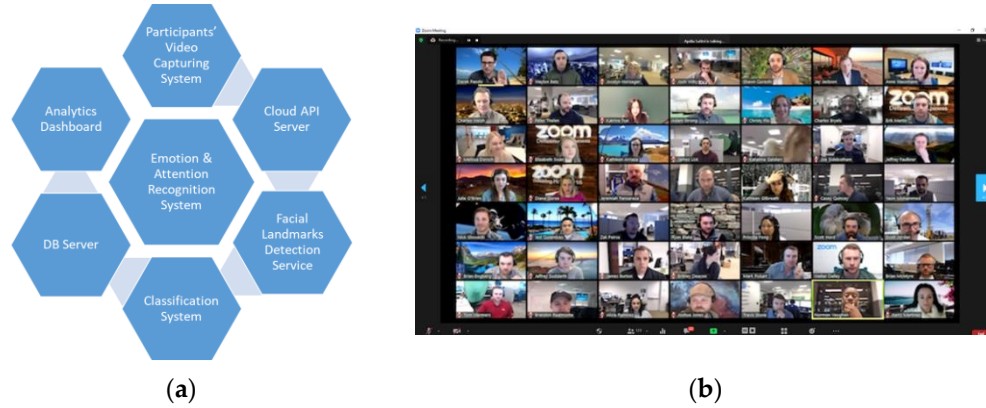

**Figure 1.** Proposed solutions for the crowded video conference: (**a**) proposed participant monitoring system design; (**b**) an overcrowded video conference (source: Google photos).

This paper describes the design of a participant engagement level detection system with a neural accelerator chip to speed up the overall process. This research also discusses emotion and attention recognition using a novel set of features to train comparatively simpler SVM models without sacrificing accuracy compared to previous similar studies.

## 2. Related Work

### 2.1. Video Conferencing Tools and Zoom

In the video conferring arena, Zoom plays a big role due to its featureful software tool and easy-to-use application interface. It provides video telephony and online chat services through a cloud-based peer-to-peer software platform. In general, teleconferencing, telecommuting, distance education, and social relation services are provided by Zoom [2]. Application developers are provided with two main software development kits (SDKs) named Meeting SDK and Video SDK [3].

### 2.2. Facial Emotion and Attention Level

There are studies in the literature about the connection between the attention level and the emotion level of a person [4,5]. In some related studies, the researcher uses electroencephalogram (EEG) signals with a brain–computer interface (BCI) to classify persons into different emotion categories [6–8].

There are widely used image-based facial emotion recognition approaches based on the facial action coding system (FACS) [9], which describes facial expressions by looking at facial muscle activation and defining action units (AUs). Examples of studies conducted in this area include calculating coefficients for facial points with Gabor wavelet filters method to extract facial appearance information [10] and Akhsay's method of calculating 21 normalized vectors to identify facial expressions [11].

When it comes to video conferencing, the participant's head pose and facial expression can give an estimate about whether the person is actively attending the video conference or not [12]. This can be easily achieved by analyzing the front-facing camera video of the participant. Real-time recognition and person-irrelevant machine learning approaches have been used to achieve this task [13].

The use of machine learning models in facial emotion detection can take many different approaches. There has been substantial research in this area of study, achieving satisfactory results in attention detection [14,15].

### 2.3. Convolutional Neural Network and Neural Accelerator Chips

Convolutional neural networks are common approaches in machine learning for highly complex neural networks which consume a significant amount of energy to perform the task. This is a challenging problem to solve for applying machine learning to embedded computers. Most of the applications require real-time results where people use edge

computing devices with machine learning abilities by consuming a minimum amount of energy [16,17].

Neural accelerator chips mainly enable this ability of embedded computers even when using a single-board computer such as Raspberry Pi. Because of the limitation on neural accelerator chips when the CNN model is picked, it has to be comfortable with the chip architecture. Most common and state-of-the-art CNNs such as AlexNet, VGGNet, and ResNet utilize fewer resources and arithmetic intensity, which is ideal when running on a neural accelerator chip [18,19].

The GTI company constructed their special CNN architecture called "GnetDet" for object detection, which can run on their AI accelerator chip named GTI Lightspeeur 2803. In addition to the object detection neural network, they have neural networks for image classification (GnetFC) and image segmentation (GnetSeg). The GTI Lightspeeur 2803 AI chip itself is very power-efficient as it can perform 24 trillion operations per second (24 TOPS) per watt, i.e., 556 frames per second (556 FPS) per watt. The input image size that can be handled by a single chip is limited to 224 × 224 pixels [20]. There are also field-programmable gate array (FPGA)-based accelerator solutions for sparse CNNs [21].

### 2.4. Dlib and OpenCV for Facial Landmark Detection

A C++ based machine learning toolkit called 'Dlib' is used by the community to create software for real-world applications. Dlib is a well-maintained open-source library with many features such as complete documentation, high-quality portable code, many built-in machine learning algorithms, and numerical algorithms. In addition to these major features, Dlib comes with GUI tools and compression algorithms. When it comes to multithreaded and networked applications, Dlib also provides decent support [22,23].

OpenCV is an open-source computer vision library. This library is also written in C and C++. This library mainly targets real-time computer vision applications. In the year 2000, the alpha version of OpenCV was released to the public, and, after many beta versions, OpenCV version 1.0 was released in 2006. Over the years, this open-source library has been improved. This computer vision library migrated to the C++ API version from the C-based API version in 2012 with their 2.4 major release. Since then, the community has been given many major releases every 6 months. OpenCV has a modularized architecture with key modules such as the core module with data structure matrix operations and basic functions (core), image processing module (imgproc), video analysis module (video), camera calibration and 3D reconstruction (calib3d), object detection module (objdetect), and many more [24,25]. These modules are especially useful in this research context to analyze video feeds and detect facial landmarks.

The facial landmark detection task can be simplified into multiple shape detection problems. Once the algorithm knows the face bounding box, i.e., the region of interest (ROI), the shape detector can detect facial structure. In order to detect a face, the algorithm must identify facial regions such as eyebrows (left and right), eyes (left and right), nose, and mouth. Dlib is a powerful facial landmark detector created using "One Millisecond Face Alignment with an Ensemble of Regression Trees" research paper written by Kazemi and Sullivan in 2014. As it sounds, this algorithm very quickly determines the facial landmarks of a human face [26–28].

### 2.5. Support Vector Machine Classification Models

Support vector machines use supervised learning methods for classification applications. In addition to classification problems, SVMs can be used for regression and outlier detection problems. Key advantages of SVMs are their memory efficiency and effectiveness in high-dimensional spaces. Furthermore, these ML models are good for sample training with high-dimensional cases. SVMs bring very few disadvantages; for example, probability estimates are not available, for which k-fold cross-validation needs to be used. Scikit-learn SVM is being very popular among the python software developer community in the data science field [29–31].

## 3. Methodology

### 3.1. System Design

The proposed system (Figure 1a) fully automates the participant's engagement level analysis process with a user-friendly interface similar to a web application. This system consists of major subsystems such as the participant video capturing system, cloud (application programming interface) API server, facial landmark detection service (utilizing the GTI AI accelerator chip to speed up the process), classification system (FER and attention level classification based on detected facial landmarks with SVMs), analytics dashboard, and database server (analytics data of each participant).

This system has a loosely coupled architecture, i.e., a modularized architecture to increase the capability of customization and future improvements. All subsystems are loosely coupled via the API interface, which enables this architecture to work smoothly. Each service runs on a different network port, allowing subsystems to talk back and forth whenever needed. A timestamp-based method is used to maintain the flow of information and trigger necessary events. Both incoming and outgoing data processes use the UNIX time-based timestamp format.

This study considers two different architectures as a function of video capturing methods. They are referred to as architecture A (Figure 2a) and architecture B (Figure 3).

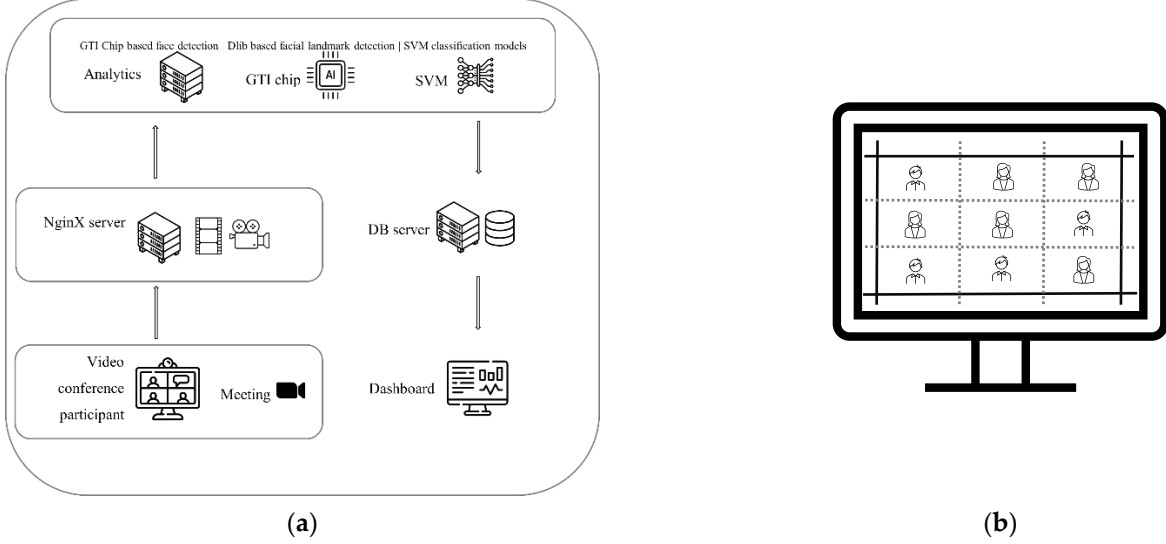

(**a**)           (**b**)

**Figure 2.** Architecture A: (**a**) system architecture; (**b**) video slicing interface.

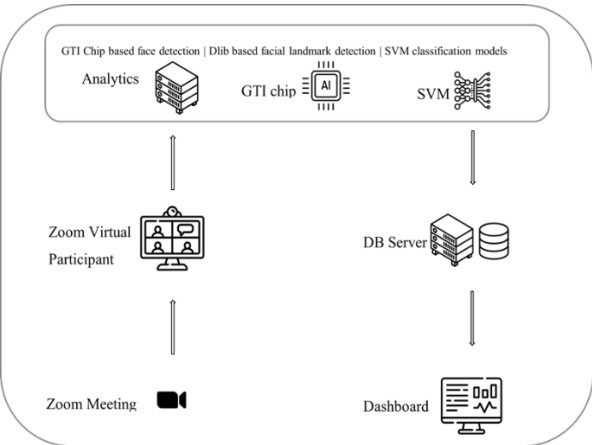

**Figure 3.** Architecture B system architecture.

### 3.1.1. Participants' Video Capturing System

This subsystem was built using the Zoom web SDK to work with any video conference conducted on the Zoom platform. By providing a meeting ID and password to this automated system, the user (meeting organizer) can enable the subsystem to join the meeting as a virtual participant, which scans the entire participant list and sends each participant's image after tagging their names. This simple virtual participant subsystem only appears in one of the implementations described in this paper (Figure 3). Another implantation (Figure 2) of this system is a comparatively more complex and less reliable alternative, especially for a bigger audience. However, the second implementation allows it to work with other video conferencing tools such as Microsoft Teams and Skype. In either method, the endpoint of the API request is the cloud API server. The received information is passed to other subsystems by the API service for further analysis. This operation is repeated every 1 min by default (this is a configurable value depending on the audience size; in this study, 500 ms was allocated per participant for capturing the camera image). A higher sampling rate will increase the quality of the analysis of the captured images.

### 3.1.2. Cloud API Services

API services help to bring all captured images to the server computer for further analysis while keeping the flow of information. This system keeps timestamps and image labels in an ordered manner throughout the processes of the entire system. The API server subsystem passes incoming data flow to the GTI chip-based subsystem to conduct analytics (facial landmark detection for every face image). Additionally, the API services help to store and retrieve meeting data from the database (DB) server while acting as the communication bridge for the entire system.

### 3.1.3. Facial Landmark Detection Service

This subsystem is the starting point of the common process for both implementations (architecture A and B) discussed in this paper. The GTI Lightspeeur 2803 chip is utilized in this subsystem to identify face bounding boxes. Then, the pretrained Dlib model is used to identify 68 facial landmarks of the detected faces one by one, as shown in Figure 4a. Finally, the calculated facial landmarks (Figure 4) are sent back to the API service for forwarding to the classification subsystem.

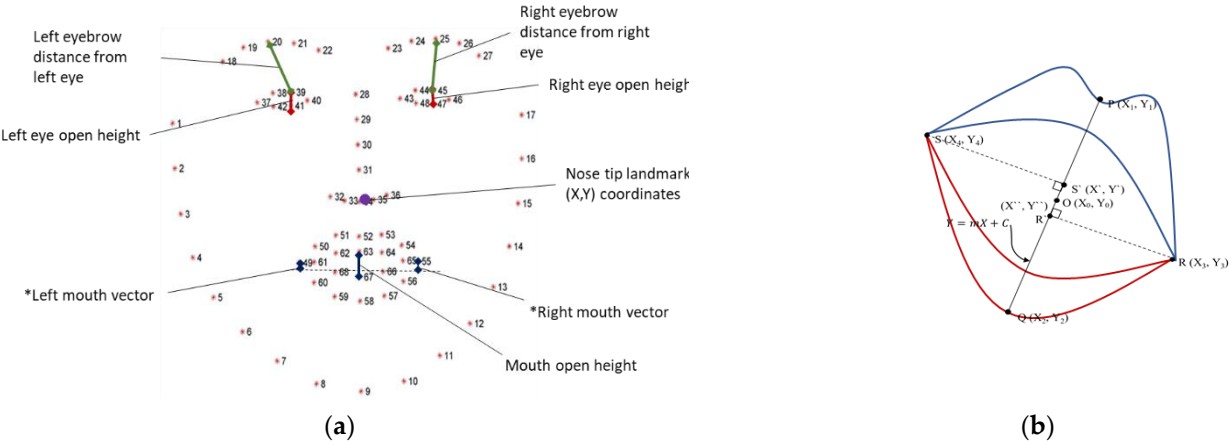

(**a**) (**b**)

**Figure 4.** The 68 facial landmark points detected by the Dlib-based model: (**a**) features used for face image classification; (**b**) how the left mouth vector (OS') and the right mouth vector (OR') calculation. * features are calculated with P, Q, R, and S points (52, 58, 55, and 49 landmark points, respectively).

### 3.1.4. Classification System

The calculated facial landmarks are used to classify faces into emotion categories and attention levels using the pretrained SVM models. The detected classes of both classifications are then sent to the DB server in each cycle via API services.

### 3.1.5. Analytics Dashboard and DB Server

The DB server stores all the required data about meetings and participants with their emotion and attention analytics data in a timeseries manner. The DB server supports all (create, read, update, and delete) CRUD operations and a few more application-specific operations via the API interface. The analytics dashboard sends HTTP POST requests to retrieve data from the DB server when needed.

The analytics dashboard has a JSON web token (JWT) authentication-based security for user access management to protect the meeting data. Users with a high enough privilege level can monitor ongoing meetings in real time or can access meeting data and generated reports for previous meetings.

### 3.2. Video Conference Monitoring Interface

In architecture A, the system can accept live screen recording or recorded video as the input. This python-based interface was developed using OpenCV to receive the captured real-time video stream through the open-source software OBS with the help of the NginX streaming server. This application is capable of splitting video frames into pieces depending on the row and column configuration given by the user. In addition to the number of columns and rows, there is an offset control feature to define the region of interest in the video frame before slicing (Figure 2b). Then, the image of each participant is sent to the server as a form submission using the API interface.

The video conference interface is one of the systems which interacts with the user directly; hence, this system needs to have a user-friendly interface. The Zoom web SDK (a version of Zoom Meeting SDK) was used to create the capturing software in design architecture B; its main task is to provide the web interface to give access to the AI tool to attend the meeting as a virtual participant. The remainder of the process is fully automated. This AI tool works as an observer, and it is capable of navigating through the entire meeting even when the participants are shown on multiple pages. The AI tool uses pagination buttons of the Zoom interface in the same way that a person would do to see all participants. This is possible because the tool can automatically create events such as hovering and button clicking by running JavaScript programs. After scanning the details and face images of all the participants in the meeting, the information is sent to the server as a form submission via an API interface.

### 3.3. Face Detection

The 'Widerface' dataset was used to train the Gnetdet CNN model using the model development kit provided by the GTI with the help of TensorFlow. Once the generation of training and testing image sets is completed with a balanced number of images in each object class (this was not the case here because this model was only trained to identify human faces, i.e., only one class), then the model training process can start and produce the model for the targeted chip.

Then, the output of the process "model" file can be used with the GTI neural accelerator chip through the software development kit provided by the GTI. This chip interfacing software is implemented in a C++ server-side application, making it a service. This service crops images using the face bounding box and then hands them over to the Dlib part of the application to mark facial landmarks easily, as discussed in the next section. This significantly increases the speed of the Dlib face landmark detection. This service keeps track of the file directory and the timestamp file. Once it detects a new set of images with the latest timestamps, the service hands over all the images to the chip in an asynchronous

manner, and, once each image is processed for face detection, the images are transferred to the Dlib part of the application.

### 3.4. Dlib Facial Landmark Detection and Preprocessing

A pretrained Dlib-based facial landmark detector was used for this research; this detector is capable of returning 68 facial landmarks of a given human face image (Figure 4a). This detector runs on the server as a service written in C++m and it can detect and record all sixty-eight landmarks under the image name in JavaScript object notation (JSON) format.

Firstly, in the preprocessing stage, all the landmarks related to detected faces are standardized into 300 px (width) × 400 px (height) after finding the size of the bounding box of the face. Therefore, all the face bounding boxes for each face image end up with the same 300 px × 400 px size. Using 68 facial landmarks, several Euclidean distances, key landmark points, and a few calculated values are obtained and used as inputs to the SVM models to identify the relevant emotion class and the attention level. The key features are shown in Table 1. This is a novel and very important dimension reduction mechanism due to the fact that it reduces 136 values related to 68 facial landmark points to 11 values without losing needed information for the classification models. This reduces the complexity of the ML models.

**Table 1.** Calculated features based on facial landmarks.

| Feature Name | Description |
| --- | --- |
| Mouth opening height | Euclidean distance from landmark 63 to landmark 67. |
| Left eye opening height | Average of Euclidean distance from landmark 38 to landmark 42 and Euclidean distance from landmark 39 to landmark 41. |
| Right eye opening height | Average of Euclidean distance from landmark 44 to landmark 48 and Euclidean distance from landmark 45 to landmark 47. |
| Left eyebrow distance from left eye | Euclidean distance from mid-point of facial landmarks 38 and 39 to facial landmark 20. |
| Right eyebrow distance from right eye | Euclidean distance from mid-point of facial landmarks 44 and 45 to facial landmark 25. |
| Nose tip landmark $X$-coordinate | Facial landmark 34's $X$-coordinate. |
| Nose tip landmark $Y$-coordinate | Facial landmark 34's $Y$-coordinate. |
| Mean $X$-coordinate | Average of the $X$-coordinate values of all facial landmarks. |
| Mean $Y$-coordinate | Average of the $Y$-coordinate values of all facial landmarks. |
| Left mouth vector | OS' vector calculation based on mouth landmark coordinates (Figures 4b and A1) |
| Right mouth vector | OR' vector calculation based on mouth landmark coordinates (Figures 4b and A1) |

The mean coordinates and the nose point of the facial landmarks give an idea of the face direction, i.e., an ideal indication of the head pose. Left and right mouth corner vector calculations take a few more steps than other calculated features. These are the lengths and direction signs (vectors) of the projections on a point on the vertical line going through the middle of the mouth (Figures 4b and A1). These vectors contain information about the shape of the mouth. All other Euclidean distances also indicate key features of the face such as eye-opening height and mouth-opening height (Figure 5).

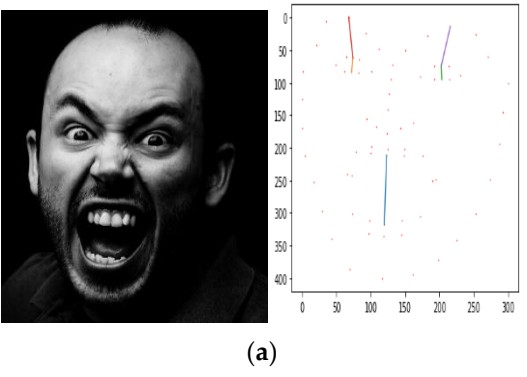 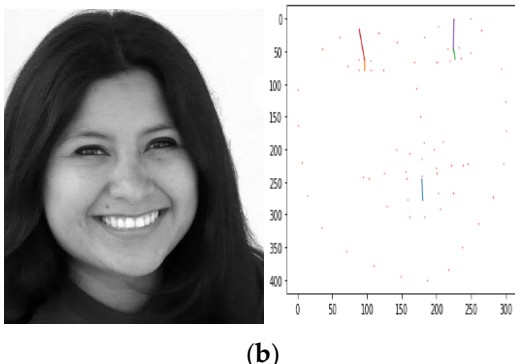

(**a**)  (**b**)

**Figure 5.** Face landmarks and feature representation of two emotions (source: Helen dataset): (**a**) "angry face" image; (**b**) "happy face" image.

After the preprocessing stage is finished for the dataset with a balanced number of images for each class, SVM model training can be started for attention classification and emotion classification. This is best practice to avoid the bias of the model toward some image classes.

### 3.5. Analytics Based on the Classification Models

In the training phase of SVM models, to identify attention level and facial emotion, the 'Helen' dataset is used. In the first step, all the facial landmarks of the detected faces of this dataset are standardized into 300 px × 400 px size at the preprocessing stage and then converted to features using the same procedure been used in the detection phase according to Table 1. Processed facial features are then used to train two models after labeling each face in the dataset with emotion class and attention level. Tables 2 and 3 show the attention level labels and emotion level labels, respectively. The 'Helen' dataset has 2000 images as training images; only 1150 images were selected considering the relevance to the study. Those selected images were labeled into five emotion classes and two attention levels by three annotators. The image annotation criteria were defined by the research team considering the research scope. Then, a defined rule set was given to the annotators for image annotation. The main rule was to focus on the head pose and the facial features of the presented person in the picture. The average emotion class and attention level for every image were used to train the models. Out of 330 validation images provided in the 'Helen' dataset, only 160 images were used for the validation of the SVM models. These trained models are used in the classification subsystem. Detected classification results are then sent to the DB server under the meeting ID and participants' names in JSON format each time.

**Table 2.** Attention classes for image labeling.

| Class Name | Description |
| --- | --- |
| Attended | Participant is paying attention to the meeting. |
| Not Attended | Participant is not paying attention to the meeting. |

**Table 3.** Emotion classes for image labeling.

| Class Name | Description |
| --- | --- |
| Excitement | Participant is showing an excited facial expression. |
| Happy | Participant is showing a happy facial expression. |
| Neutral | Participant is showing a neutral facial expression. |
| Sleepy | Participant is showing a sleepy facial expression. |
| Disgust | Participant is showing a disgusted facial expression. |

*3.6. Dashboard*

This web-based dashboard runs on a cloud server allowing anyone to access it from anywhere in the world over the internet. JavaScript is the main language used for this part of the application, which is managed by Node.js and in-page JavaScript. HTML and CSS are also used to create interfaces for a good user experience.

The dashboard is a featureful interface that gives a final summary of each meeting that monitored with the system. By retrieving data from the database using the API interface, the dashboard provides insightful analytics about the meetings as shown in Figures 6 and 7. This interface also enables users to log in to their accounts and start monitoring the meeting, as well as to access reports of the previous meeting participants available in the database.

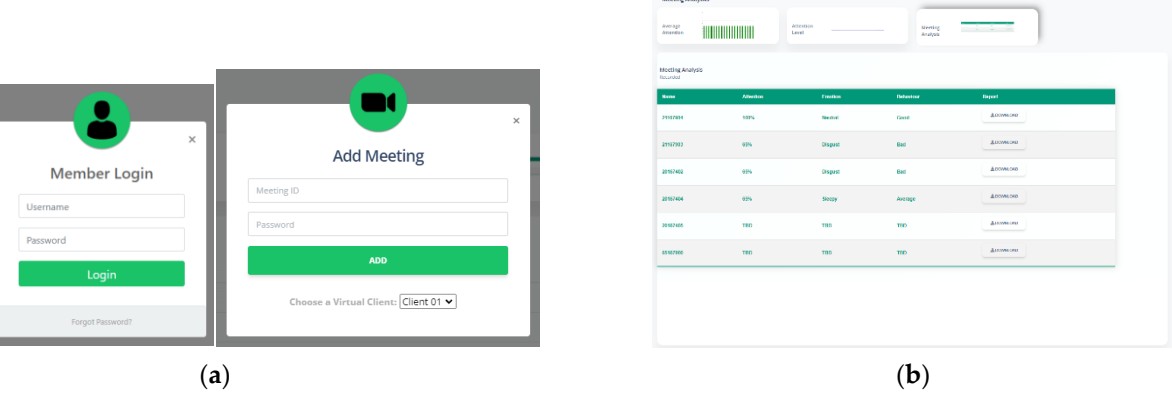

(**a**)                                                  (**b**)

**Figure 6.** Dashboard views: (**a**) login and add meeting windows; (**b**) meeting analysis table view.

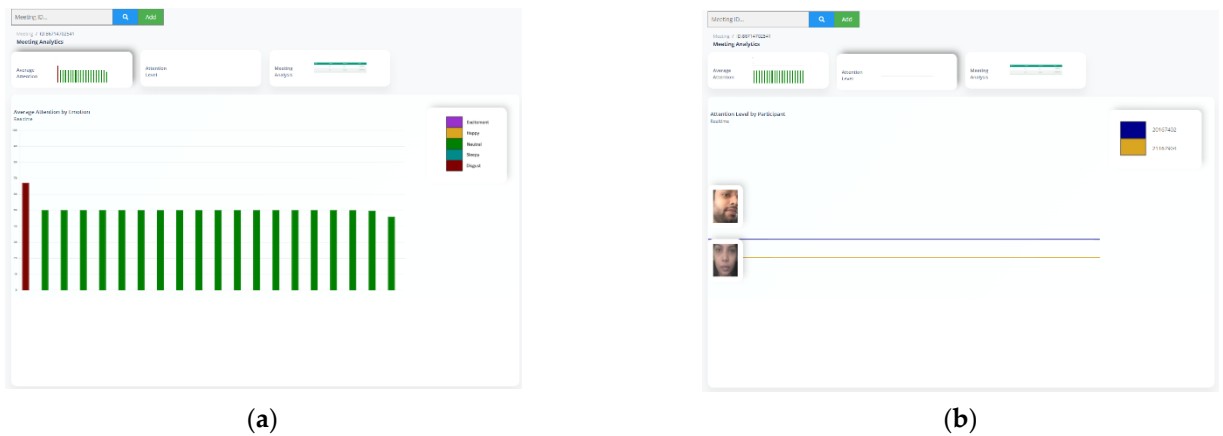

(**a**)                                                  (**b**)

**Figure 7.** Dashboard views: (**a**) average attention by emotion view; (**b**) attention level by participant view.

**4. Results**

*4.1. Analytics Results Based on the Classification Models*

According to the emotion and attention classification data, there are two major analytic types provided for participants of the zoom meeting, as shown in Figure 7a,b: a bar graph with its height based on the average attention level (Table 2) and color based on the average emotion level (Table 3) for the entire audience (Figure 7a), which is updated every 1 min (default update time of the system) with live data when the meeting is ongoing; the per-participant attention level (Table 2) graph (Figure 7b), which allows the user to observe a selected number of participants (e.g., two participants) continuously throughout the meeting. This also supports the real-time observation of an ongoing meeting. To retrieve

these analytics of already finished meetings, the user needs to search by meeting ID to obtain the relevant graph.

### 4.2. Accuracy of the Classification Models

The accuracy of the SVM classification model for emotion identification is shown in Figure 8b confusion matrix. The excited, happy, neutral, sleepy, and disgusted emotions achieved 65%, 60%, 61%, 69%, and 55% accuracy of detection, respectively. The false-positive value of each emotion class was always less than 19%, except for 21% and 26% when detecting a disgusted emotion as neutral and when detecting a happy emotion as excited, respectively. This result would not greatly affect the accuracy of the final result.

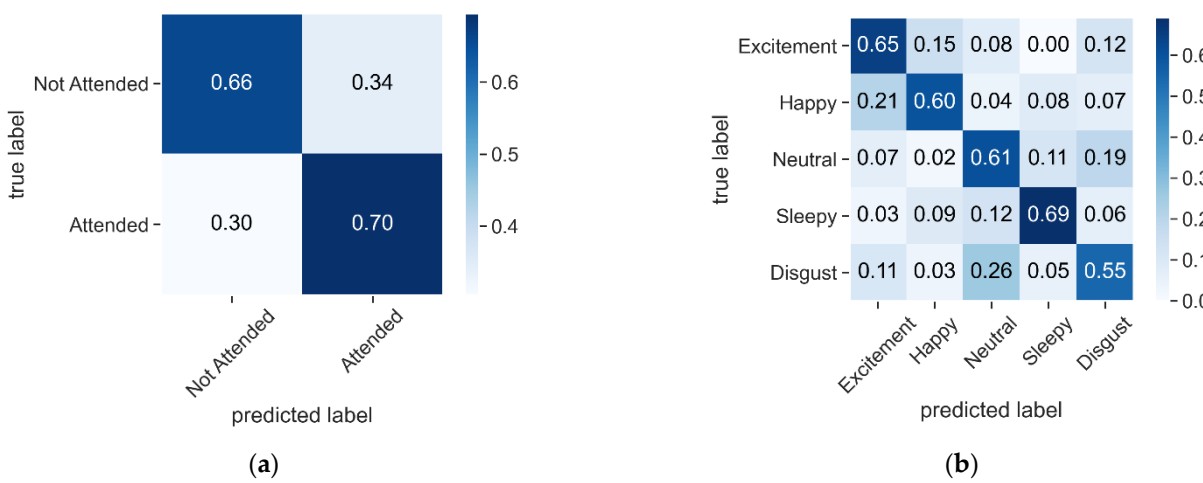

(a)                                      (b)

**Figure 8.** Confusion matrices for detection SVM models (10-fold cross-validation results): (**a**) attention classification model; (**b**) face emotion classification model.

The attention level to the meeting was classified into two states as shown in Table 2. After training with the balanced image set for attention classification, the accuracy of the attention model was evaluated using the confusion matrix in Figure 8a. When the participant was actively engaging in the video conference, a 30% true-negative level was achieved; thus, 70% of actively engaged participants could be accurately detected. The false-positive rate was 34%; thus, 66% of participants were correctly identified as not engaged in the meeting. This model had an F1 score of 0.6725 and accuracy of 0.6770.

### 4.3. Report Generation for Participants

The dashboard has a tab view (Figure 6b) where the user can see a summary of all participants' performances in a given meeting in a table format, with their name (the name used for the video call), attention level, behavior (considering the emotion attention matrix in Table 4), and there is an option to download the report, as shown in Figure 9.

**Table 4.** Behavior matrix based on emotion class and attention level.

|  | Excitement | Happy | Neutral | Sleepy | Disgust |
|---|---|---|---|---|---|
| Attention | Good | Good | Good | Average | Bad |
| No attention | Good | Good | Average | Bad | Bad |

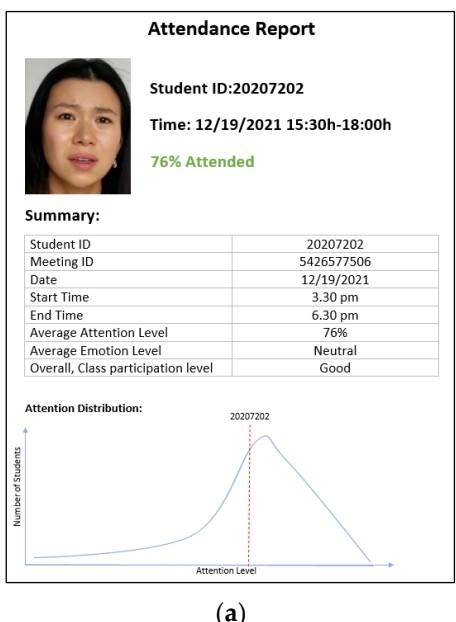

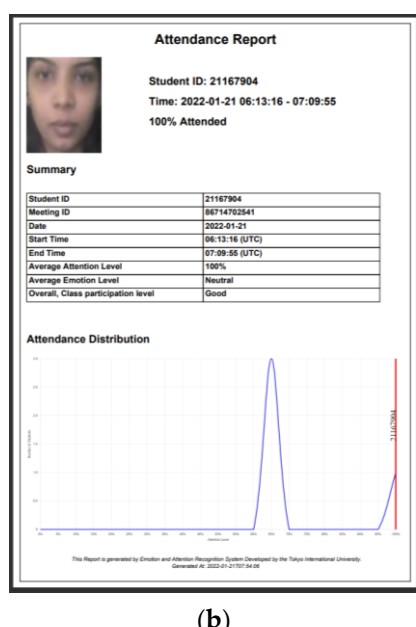

(**a**)          (**b**)

**Figure 9.** Attendance report generated by the system for a video conference: (**a**) theoretical report generated for a bigger audience; (**b**) actual report generated for four–participant audience.

The report consists of four major elements: image of the participant (captured image by the system), display name of the participant, meeting time and attendance percentage, and a summary of the meeting attention distribution of participants with respect to the entire audience. For a big enough audience size, this attention distribution should present a bell-shaped curve with negative skewness if the video conference is able to grasp the attention of the audience. In the opposite scenario, the curve would show positive skewness (Figure 9a). In the case of a smaller audience with good general attention to the video conference, the attention distribution would present the curve shown in Figure 9b.

## 5. Discussion

When it comes to computer-based emotion recognition, EEG-based methods are more accurate; however, unlike image-based methods, the subject has to wear EEG probes and other electrical equipment, as well as be in a controlled environment. On the other hand, image-based methods are more convenient to detect the emotions of participants in a video conference, even though their accuracy is lower.

The resulting application of this research was the creation of a platform that video meeting organizers can use to monitor their audience. Architecture B is a fully automated application that Zoom meeting organizers can use to monitor a video conference (a Zoom meeting) without any hassle compared to architecture A, as shown in Figure A2. While conducting the research, two reliable SVM models were created for facial landmark-based emotion classification and attention detection. The dimension reduction method introduced in the preprocessing phase showed a significant level of improvement through a reduction of 136 features (136 dimensions) to 11 features.

Instead of using the Gabor wavelet facial appearance information method [22] or Akhsay's method of calculating 21 normalized vectors for face emotion recognition [21], this research used only 11 calculated values with a relatively lower computational cost.

This attention classification model showed 0.6770 accuracy using only 1150 images from the Helen dataset, which can be further improved with more training samples. The loosely coupled architecture design allows continuous improvement of the classification subsystem without disrupting other modules. The image quality of the captured image is a particularly important factor for classification accuracy. Therefore, future improvement of the image capturing subsystem can significantly enhance the effectiveness of the entire



system. Using the Zoom video SDK or a video conferencing tool with direct participant video feed access can noticeably improve the result of the overall system.

When using this system in schools, the attendance of students can also be achieved automatically.

A public dataset was used in this research for training the model. For system testing, a group of students were selected after explaining the terms of use of their personal data. This experiment did not record images but instead kept timeseries data for each student with respect to their attention and emotion classification. When this system is converted to a production-level application, the EU General Data Protection Regulation (GDPR) needs to be implemented, and the terms of service should clearly explain the data collection, access, use, and processing of personal data in terms of the legal basis and purpose of use related to the retention period, data storage and transfer, prior consent from the person, and personal rights.

## 6. Conclusions

It has been proven that AI and machine learning applications can make our life easier by automating human operations with the help of artificial systems. Even though the system described in this paper can be further improved, it showed reliable results and accuracy levels in detecting human faces, emotion classes, and attention levels of participants in a meeting. This is not an easy task for a human operator, even with a lower observation rate. Furthermore, this AI tool works in the microsecond time range compared to the second time range of humans in most cases.

Most importantly, a human operator cannot match the classification speed and repeatability of artificial systems. By optimizing the classification models of this proposed system, broader applications can be achieved.

**Author Contributions:** J.K., system architecture, algorithm development, software development of backend, frontend, and API server of the proposed system, and validation of results; D.D.A., paper reviewing and validation of results; J.R., guidance and research advice. All authors have read and agreed to the published version of the manuscript.

**Funding:** This research received no external funding.

**Institutional Review Board Statement:** Not applicable.

**Informed Consent Statement:** Not applicable.

**Data Availability Statement:** Not applicable.

**Acknowledgments:** Gyrfalcon Technologies Inc (GTI) and its Japan team are thanked for the great cooperation and provision of their GTI Lighspeeur 2803 for testing our algorithm.

**Conflicts of Interest:** The authors declare no conflict of interest.

**Appendix A**

$$X_0 = \frac{(X_1 + X_2)}{2}; Y_0 = \frac{(Y_1 + Y_2)}{2}$$

$If \ X_1 \neq X_2;$
$$m = \frac{(Y_1 - Y_2)}{(X_1 - X_2)}$$

$$\frac{(Y_4 - Y')}{(X_4 - X')} \times m = -1$$

$$mY' + X' = mY_4 + X_4 \quad \longleftarrow \quad ①$$

$$\frac{(Y_2 - Y')}{(X_2 - X')} = m$$

$$Y' - mX' = Y_2 - mX_2 \quad \longleftarrow \quad ②$$

$$m^2Y' + Y' = m^2Y_4 + mX_4 + Y_4 - mX_2; \qquad ① \times m + ②$$

$$Y' = \frac{m^2Y_4 + mX_4 + Y_2 - mX_2}{m^2 + 1}$$

$$X' = \frac{mY_4 + X_4 - mY_2 - m^2X_2}{m^2 + 1}$$

$If \ X_1 = X_2;$

$$Y' = Y_4$$
$$X' = X_1 = X_2$$

$Similarly \ ; \ If \ X_1 \neq X_2;$

$$m = \frac{(Y_1 - Y_2)}{(X_1 - X_2)}$$

$$Y'' = \frac{m^2Y_3 + mX_3 + Y_2 - mX_2}{m^2 + 1}$$

$$X'' = \frac{mY_3 + X_3 - mY_2 - m^2X_2}{m^2 + 1}$$

$If \ X_1 = X_2;$

$$Y'' = Y_3$$
$$X'' = X_1 = X_2$$

**Figure A1.** Vector calculation based on mouth landmark coordinates.

**Appendix B**

**Figure A2.** User journey of proposed system (automated system implementation).

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
