# Peer review of "An Emotion and Attention Recognition System to Classify the Level of Engagement to a Video Conversation by Participants in Real Time Using Machine Learning Models and Utilizing a Neural Accelerator Chip"

_algorithms, doi:10.3390/a15050150_

Round 1

Reviewer 1 Report

This manuscript has exposed an approach for detecting the level of engagement in a video conversation by participants in real-time using machine learning techniques.

This study sounds very interesting and very useful for many people in our society. However, some weaknesses must be fixed before considering the manuscript for publication.

Major changes:

1. The introduction is mixed with a bit of state of the art. Authors must separate the introduction in these two sections and state of the art must be extended.

2. Although implementing a prediction model into software is essential, this fact is not so important for scientific purposes. Due to it, the manuscript must be more focused on explaining the advance in relation to the predictive models. Authors must explain how they can do the preprocessing of data, training, validation, etc.

3. It is important to clarify how they consider if a person is very engaged or not to train an ML model.

4. The discussion of the manuscript should be between their tool or models and other existing models in the literature. Why your ML model is better than others?

5. Authors need to share the models and tools with the rest of the world.

I think it's a very interesting tool, but it is difficult for me to see the scientific advance.

Reviewer 2 Report

The authors have well studied the requirement for the study and well presented the methods and application. The study has a good flow without any edits. However, I strongly suggest authors demonstrate their results to convince and get attention of the readers. 

Reviewer 3 Report

The paper is very interesting. However, I have a few comments.

Data protection (GDPR) is now required.  In the paper, the authors do not indicate how this problem resolved. It is necessary to inform the participants during the experiment that their faces will be scanned (captured). This is very sensitive data.

In the paper, the authors used only 17 references. I recommend to increase the number of references. Please, focus especially on the issue of emotions. I missed in the paper according to which model (psychological) the authors classified emotions.

Existing several models (Plutchick, Russell, Ekman...) but the way of classifying emotions is different. For example, Ekman's classification is used when looking directly (frontally) at the participant's face. However, it does not include the sort of emotions that the authors of the paper have classified. It is therefore important to state what model the authors used.

Round 2

Reviewer 1 Report

I thank the authors for taking all my suggestions on board. However, there are still some aspects that need to be remedied:

1. The introduction still contains subsections. The authors should include the subsections of the introduction in a new section called Related Work so that there is a clear distinction between the introduction and the state of the art of the problem.

2. They indicate that they will share everything on a public github. This is interesting, but for us reviewers to be able to assess the veracity of what they indicate in the manuscript, we need you to send us the documents if you do not want to share it already on github. Otherwise, we will not be able to verify that what you say you have done is correct.

Round 3

Reviewer 1 Report

The manuscript has improved sufficiently to be considered for publication in this journal.